# Ectopic Activation of *Fgf8* in Dental Mesenchyme Causes Incisor Agenesis and Molar Microdontia

**DOI:** 10.3390/ijms25137045

**Published:** 2024-06-27

**Authors:** Yu Wang, Jingjing Wang, Tian Xu, Shuhui Yang, Xinran Wang, Lei Zhu, Nan Li, Bo Liu, Jing Xiao, Chao Liu

**Affiliations:** 1Institute of Stomatology, Binzhou Medical University, Yantai 264003, China; wy15098567078@bzmc.edu.cn; 2Department of Oral Pathology, School of Stomatology, Dalian Medical University, Dalian 116044, China; wangjj847@163.com (J.W.); 18956359791@163.com (T.X.); yyyysh974@163.com (S.Y.); 18999151792@163.com (X.W.); zhul03@dmu.edu.cn (L.Z.); linandy@dmu.edu.cn (N.L.); 3Institute for Genome Engineered Animal Models of Human Diseases, Dalian Medical University, Dalian 116044, China; liub03@dmu.edu.cn

**Keywords:** biomineralization, dental mesenchyme, *Fgf8*, FGF signaling, microdontia, odontoblast differentiation, *Shh*, tooth development, tooth agenesis

## Abstract

Putatively, tooth agenesis was attributed to the initiation failure of tooth germs, though little is known about the histological and molecular alterations. To address if constitutively active FGF signaling is associated with tooth agenesis, we activated *Fgf8* in dental mesenchyme with *Osr-cre* knock-in allele in mice (*Osr2-cre^KI^*; *Rosa26R-Fgf8*) and found incisor agenesis and molar microdontia. The cell survival assay showed tremendous apoptosis in both the *Osr2-cre^KI^*; *Rosa26R-Fgf8* incisor epithelium and mesenchyme, which initiated incisor regression from cap stage. In situ hybridization displayed vanished *Shh* transcription, and immunostaining exhibited reduced Runx2 expression and enlarged mesenchymal Lef1 domain in *Osr2-cre^KI^*; *Rosa26R-Fgf8* incisors, both of which were suggested to enhance apoptosis. In contrast, *Osr2-cre^KI^*; *Rosa26R-Fgf8* molar germs displayed mildly suppressed *Shh* transcription, and the increased expression of *Ectodin*, Runx2 and Lef1. Although mildly smaller than WT controls prenatally, the *Osr2-cre^KI^*; *Rosa26R-Fgf8* molar germs produced a miniature tooth with impaired mineralization after a 6-week sub-renal culture. Intriguingly, the implanted *Osr2-cre^KI^*; *Rosa26R-Fgf8* molar germs exhibited delayed odontoblast differentiation and accelerated ameloblast maturation. Collectively, the ectopically activated *Fgf8* in dental mesenchyme caused incisor agenesis by triggering incisor regression and postnatal molar microdontia. Our findings reported tooth agenesis resulting from the regression from the early bell stage and implicated a correlation between tooth agenesis and microdontia.

## 1. Introduction

Tooth development relies on the intensive interactions between the dental epithelium and the underlying mesenchyme, in which a lot of growth factors, such as Wnts, Transformation Growth Factor-β (TGF-β)/Bone Morphogenic Protein (BMP), Sonic Hedgehog (SHH) and Fibroblast Growth Factor (FGF) families, are involved [1]. The FGF family consists of 22 members, which are divided into seven groups (Fgf1, Fgf4, Fgf7, Fgf8, Fgf9, Fgf11 and Fgf15 groups) according to their affinities to FGF receptor 1–4 (Fgfr1–4) [2]. In the past decades, a lot of studies revealed the roles of FGF signaling in the patterning, initiation, morphogenesis and differentiation of tooth development [3]. During early tooth development, *Fgf8*-expressing epithelial progenitors in mandible migrated toward the *Shh*-expressing domain to decide the position of tooth primordium. Suppression of FGF and SHH signaling resulted in tooth agenesis by disrupting the oriented cell migration of cells [4]. In contrast, the deletion of *Sprouty2* (*Spry2*) and *Sprouty4* (*Spry4*), the intracellular suppressor of FGF signaling, led to supernumerary teeth by persistently activating FGF signaling [5,6]. During tooth morphogenesis, the expressions of *Fgf4*, *Fgf9* and *Fgf20* in dental epithelium acted as the major downstream effectors of ectodysplasin (Eda) that could rescue the reduced size, number and shape of teeth resulting from the loss-of-function of *Eda* [7]. In contrast to the epithelium-expressed *Fgf8*, *Fgf4*, *Fgf9* and *Fgf20*, *Fgf3* and *Fgf10* were activated in dental mesenchyme from E12.5 and maintained by *Runt Related Transcription Factor 2* (*Runx2*) to form interaction loop with the epithelial *Fgf4* and *Fgf10*. *Runx2* null mice showed tooth germs arrested at early cap stage with the reduced *Fgf3*, *Fgf10*, *Fgf4* and *Fgf9* expression [8]. Moreover, the *Fgf10* expressed in muse incisor mesenchyme adjacent to the cervical loop maintained the stem cell niche for the self-renewal of ameloblasts and odontoblasts [9]. The incisor germs in *Fgf10* null mice proceeded to cap stage without the formation of a cervical loop because of tremendous cell death [10]. Therefore, disruptions in the expression of *Fgfs*, as well as the FGF signaling in dental epithelial and/or mesenchymal cells, could lead to the developmental failure of tooth germs, which causes tooth agenesis in clinics.

Tooth agenesis is categorized into hypodontia, referring to the absence of six or less teeth, and oligodontia, referring to the lack of more than six teeth [11]. Genes involved in tooth agenesis have been identified in humans, such as *Paired Homeobox Gene 9* (*PAX9*), *Muscle Segment Homeobox Homolog 1* (*MSX1*), *Axis Inhibition Protein* 2 (*AXIN2*), *WNT10A*, *WNT10B*, *EDA* and *EDAR*, and verified in genetic mouse models [12]. A series of studies indicated that the mesenchymal Wnt signaling was modulated by the antagonistic interaction between Axin2 and Runx2, and an increased Wnt signaling in dental mesenchyme impaired tooth development [13]. Mutations of *AXIN2*, an inhibitor to Wnt signaling, are associated with tooth agenesis [14], while the haplo-insufficiency of *RUNX2* results in supernumerary teeth [15]; thus, the exact mechanism of *Runx2* in tooth development remains to be elucidated. On the other hand, although the roles of these genes been intensively studied during tooth development, few investigations have concerned the pathogenesis of tooth agenesis. Since few studies have reported on the degeneration of tooth germs in the late cap or bell stage, tooth agenesis is putatively attributed to the failure of tooth initiation, or the arrest at the bud or early cap stage [12,16].

Compared to tooth agenesis, microdontia, which is characterized by the miniature teeth, is relatively rare. However, the definition of microdontia is still in debate, because microdontia is usually companied by micrognathia and regarded as a complication of micrognathia. Actually, regardless of micrognathia, the etiology of microdontia also remains elusive. Most microdontia was reported as a complication in syndromes such as LAMM syndrome (labyrinthine aplasia, microtia and microdontia), Axenfeld–Rieger syndrome, Lacrimo-auriculo-dento-digital syndrome, Kenny–Caffey Syndrome, etc. [17,18,19,20]. Because of the mutations detected in above syndromes, *FGF3*, *FGF10*, *FGFR2* and *FGFR3*, especially *FGF3*, are candidates for the causative gene of microdontia [17,19,21]. Since tooth eruption takes place after birth, tooth agenesis or microdontia can only be found in childhood, and that impeded our understanding of their pathological processes.

As mentioned above, FGF signaling is involved in both tooth agenesis and microdontia. However, the exact roles of FGF signaling in tooth agenesis and microdontia are still being debated, which requires both the loss- and gain-of-function models for further exploration. Since most of the mutations reduce gene expression or protein activity, previous reports on the mutations of *FGF* genes or FGF signaling could be regarded as the model of loss-of-function [22]. By contrast, there are a few reports on the gain-of-function models to clarify the roles of FGF genes or signaling in tooth agenesis and microdontia. In this study, by activating *Fgf8* in dental mesenchyme, we constructed a gain-of-function model of FGF signaling. Since the attenuated FGF signaling in dental mesenchyme is correlated with the reduced tooth number, we hypothesized that *Osr2-cre^KI^; Rosa26R-Fgf8* mice would give rise to supernumerary tooth as *RUNX2* haplo-insufficient cases did [15]. However, we found the agenesis of incisors and the microdontia in molars. Our study not only revealed a precise pathogenesis of tooth agenesis but also suggested a potential correlation between tooth agenesis and microdontia.

## 2. Results

### 2.1. The Regressed Incisor Germs in Osr2-cre^KI^; Rosa26R-Fgf8 Mice

To explore the effects of mesenchyme-derived Fgf8 on incisor and molar development, we examined the histological characteristics of *Osr2-cre^KI^*; *Rosa26R-Fgf8* tooth germs. In the bud stage, both the E12.5 and E13.5 WT incisor (Figure 1A,C) and molar germs (Figure 1B,D) showed little difference from those in the E12.5 and E13.5 *Osr2-cre^KI^; Rosa26R-Fgf8* incisor (Figure 1A′,C′) and molar germs (Figure 1B′,D′). In the caps stage, the E14.5 WT incisor germs invaginated into mesenchyme deeply (Figure 1E), while the *Osr2-cre^KI^*; *Rosa26R-Fgf8* incisor germs still kept the epithelial stalk connecting the cap-like enamel organ to oral epithelium, suggesting an insufficient invagination of epithelial compartment (Figure 1E′). Meanwhile, the *Osr2-cre^KI^*; *Rosa26R-Fgf8* molar germs (Figure 1F′) also displayed the smaller and shallower invaginated cap-like enamel organs compared to the WT molar germs (Figure 1F). In the early bell stage, the epithelial stalks of E16.5 *Osr2-cre^KI^*; *Rosa26R-Fgf8* incisor germs still remained (Figure 1G′), in contrast to the WT incisor enamel organs completely separated from oral epithelium (Figure 1G). Moreover, there was no discernible secondary enamel knot (EK) and cusp in *Osr2-cre^KI^*; *Rosa26R-Fgf8* molar germs (Figure 1H′) compared with the WT controls (Figure 1H). Worthy of noticing, the elongated and polarized pre-odontoblasts and ameloblasts were obvious in WT incisor and molar germs (Figure 1G,H) but still blurred in the *Osr2-cre^KI^*; *Rosa26R-Fgf8* incisor and molar germs (Figure 1G′,H′). Most intriguingly, the E18.5 *Osr2-cre^KI^*; *Rosa26R-Fgf8* incisor germs disappeared completely (Figure 1I′), while the molar germs were only smaller with the blurred secondary EKs (Figure 1J′) than the WT controls (Figure 1I,J). These findings indicated that ectopic activation of *Fgf8* in dental mesenchyme led to the degeneration of incisor germs but suppressed molar development more mildly.

### 2.2. Enhanced Cell Apoptosis in Osr2-cre^KI^;Rosa26R-Fgf8 Incisor Germs

To address the obviously regressed incisor germs and the reduced size of molar germs in *Osr2-cre^KI^*; *Rosa26R-Fgf8* mice, the cell proliferation and apoptosis were examined by BrdU and TUNEL assays. In the E13.5 *Osr2-cre^KI^*; *Rosa26R-Fgf8* incisor germs, the density of the BrdU-positive nuclei in epithelium was comparable to that of the WT control, but it was higher in mesenchyme than that of the WT control (Figure 2A,A′,C). In contrast, in the E13.5 *Osr2-cre^KI^*; *Rosa26R-Fgf8* molar germs, the density of BrdU-positive nuclei was lower in epithelium but higher in mesenchyme compared to the WT counterparts (Figure 2B,B′,C). Surprisingly, both the E14.5 *Osr2-cre^KI^*; *Rosa26R-Fgf8* incisor and molar germs showed the remarkably increased cell proliferation in enamel organs but also the decreased cell proliferation in dental papilla (Figure 2D–F). On the other hand, the TUNEL assay showed no difference in cell apoptosis between both the incisor and molar germs of E14.5 WT and *Osr2-cre^KI^*; *Rosa26R-Fgf8* mice (Figure 2G,H,G′,H′). In contrast, the TUNEL signals were obviously in the enamel organs of E16.5 *Osr2-cre^KI^*; *Rosa26R-Fgf8* incisor germs, but they were almost devoid in WT counterparts (Figure 2I,I′). The *Osr2-cre^KI^*; *Rosa26R-Fgf8* molar germs differed little in TUNEL signals from the WT group (Figure 2J,J′). These results suggested that the regression of *Osr2-cre^KI^*; *Rosa26R-Fgf8* incisor germs was mainly attributed to the increased cell apoptosis in E16.5 enamel organs.

### 2.3. Differential Gene Expression between Osr2-cre^KI^; Rosa26R-Fgf8 Incisor and Molar Germs

To address the apoptosis in the enamel organs of *Osr2-cre^KI^; Rosa26R-Fgf8* incisors, the transcription of *Shh*, a growth factor expressed in dental epithelial cells key to cell proliferation, survival and differentiation, was examined by in situ hybridization. When robust in the E14.5 EKs of WT incisors, the *Shh* transcription was excluded in the enamel organs of *Osr2-cre^KI^*; *Rosa26R-Fgf8* incisors (Figure 3A,B). Similarly, the *Shh* transcription confined to the differentiating ameloblasts of E16.5 WT incisors was also absent from the epithelial compartments of *Osr2-cre^KI^*; *Rosa26R-Fgf8* incisors (Figure 3C,D). In contrast to the absence in the incisors, the *Shh*-expressing EKs were obvious in the E14.5 *Osr2-cre^KI^*; *Rosa26R-Fgf8* molars, though smaller and more concentrated to the buccal sides compared to WT molars (Figure 3E,F). At E16.5, *Shh* transcription extended into the entire inner ameloblast layer in the *Osr2-cre^KI^*; *Rosa26R-Fgf8* molars, as in WT molars, but the cusps of the *Osr2-cre^KI^*; *Rosa26R-Fgf8* molars were still inconspicuous (Figure 3G,H). Similarly, Runx2, the transcription factor mediating *Fgf3* and *Fgf10* activation in dental mesenchyme, was almost diminished in E14.5 *Osr2-cre^KI^*; *Rosa26R-Fgf8* incisor mesenchyme compared to the robust expression in WT controls (Figure 3I,J), while the Runx2 expression in E14.5 *Osr2-cre^KI^*; *Rosa26R-Fgf8* molar mesenchyme was comparable to, or even a little robuster than that in WT molars (Figure 3K,L). The transcription of *Ectodin*, a dual inhibitor to Wnt and BMP signaling, was remarkably enhanced in both the E14.5 *Osr2-cre^KI^*; *Rosa26R-Fgf8* incisor and molar epithelium compared to the WT counterparts (Figure 3M–P). Taken together, the suppressed *Shh* and Runx2 expression specific to *Osr2-cre^KI^*; *Rosa26R-Fgf8* incisor germs indicated an attenuated FGF signaling that resulted in the incisor agenesis. Meanwhile, the enhanced *Ectodin* transcription, implicating an inhibition of Wnt and BMP signaling in both *Osr2-cre^KI^*; *Rosa26R-Fgf8* incisor and molar germs, was suggested to indiscriminately repress incisor and molar development.

### 2.4. Altered Gene Expression in Osr2-cre^KI^; Rosa26R-Fgf8 Incisor and Molar Germs

Then, we examined several key signaling pathways in the E13.5 incisor and molar germs. Compared with the WT controls (Figure 4A,B), Fgfr1, the main receptor for Fgf8, was also ectopically activated in incisor enamel organs (Figure 4A′), and drastically up-regulated in the presumptive molar papilla of *Osr2-cre^KI^*; *Rosa26R-Fgf8* mice (Figure 4B′). The p-Erk1/2 distribution was sporadic in the E13.5 WT incisor mesenchyme beside the epithelial stalk (Figure 4C), but it was more intensive in the counterpart of *Osr2-cre^KI^*; *Rosa26R-Fgf8* incisor germs (Figure 4C′). In the molar mesenchyme, the intensive p-Erk1/2 signal in the WT molar papilla (Figure 4D) was almost diminished in the *Osr2-cre^KI^*; *Rosa26R-Fgf8* molar papilla, but it was ectopically activated in the mesenchyme lingual to epithelial stalks (Figure 4D′). Lef1, a nuclear mediator of canonical Wnt signaling, which was distributed in the epithelium and mesenchyme of WT incisor and molar germs (Figure 4E,F), became enlarged in the incisor mesenchyme (Figure 4E′), but fainter in the molar mesenchyme of *Osr2-cre^KI^*; *Rosa26R-Fgf8* mice (Figure 4F′), suggesting an up- and down-regulated Wnt signaling in incisor and molar mesenchyme, respectively. Compared to WT tooth germs (Figure 4G,H), the distribution of p-Smad1/5/8 seemed little difference in both the incisor mesenchyme (Figure 4G′) and molar mesenchyme of *Osr2-cre^KI^*; *Rosa26R-Fgf8* mice (Figure 4H′).

### 2.5. Delayed Odontoblast Differentiation and Enhanced Ameloblast Degeneration in Osr2-cre^KI^; Rosa26R-Fgf8 Molars

Then, we examined the ameloblastic and odontoblastic differentiation in *Osr2-cre^KI^*; *Rosa26R-Fgf8* molars by in situ hybridization. *Amelogenin*, the matrix essential for enamel formation and dominantly expressed in ameloblasts, showed no difference between WT (Figure 5A) and *Osr2-cre^KI^*; *Rosa26R-Fgf8* molars (Figure 5B) after 3 days of sub-renal culture. In contrast, *Dentin sialophosphoprotein* (*DSPP*), encoding the dentin-specific protein and activated in both the WT molar ameloblasts and odontoblasts, was detected only in the E17.5 *Osr2-cre^KI^*; *Rosa26R-Fgf8* ameloblasts but absent in the odontoblasts after 3 days of sub-renal culture (Figure 5C). However, after 5 days of sub-renal culture, although still expressed in both the E17.5 WT molar ameloblasts and odontoblasts (Figure 5E), the *DSPP* expression was only in the E17.5 *Osr2-cre^KI^*; *Rosa26R-Fgf8* molar odontoblasts and vanished from the ameloblasts (Figure 5F). Thus, it is suggested that the persistent activated *Fgf8* in molar mesenchyme delayed odontoblast differentiation but enhanced the degeneration of ameloblasts.

### 2.6. Impaired Mineralization and Reduced Size of Osr2-cre^KI^; Rosa26R-Fgf8 Molars

Since *Osr2-cre^KI^; Rosa26R-Fgf8* mice died immediately after birth, we transplanted the E17.5 *Osr2-cre^KI^; Rosa26R-Fgf8* molar germs into a sub-renal capsule for a 3- or 6-week culture to evaluate the in vivo mineralization. After 3 weeks of sub-renal culture, the X-ray images of the grafted E17.5 *Osr2-cre^KI^*; *Rosa26R-Fgf8* molar germs (Figure 6B) were similar to those of the grafted WT molar germs (Figure 6A). However, histological sections revealed that the predentin layer in the grafted *Osr2-cre^KI^*; *Rosa26R-Fgf8* molar germs (Figure 6D,D′) was much thicker than that in WT molar grafts (Figure 6C,C′). After 6 weeks of sub-renal culture, the grafted E17.5 WT molar germs gave rise to dental root, crown and alveolar bone (Figure 6E). In contrast, the grafted E17.5 *Osr2-cre^KI^*; *Rosa26R-Fgf8* molar germs only formed atypical tooth morphology and hypomineralized tooth miniatures (Figure 6F), implicating the impaired mineralization, especially in dentin, and the microdontia resulting from the dental mesenchyme-derived Fgf8. Combined with the prenatally regressed incisor germs, the persistent *Fgf8* expression in dental mesenchyme was suggested to impair not only tooth development by repressing *Shh* transcription and FGF signaling but also postnatal tooth size, root formation and dentin mineralization.

## 3. Discussion

Tooth agenesis is putatively regarded as a consequence of failed tooth initiation, because most mouse models of tooth agenesis showed tooth development arrested at the laminar or bud stage, except for *Runx2* null mice at the early cap stage [8,10,11]. In this study, we reported the incisor agenesis in *Osr2-cre^KI^*; *Rosa26R-Fgf8* mice in which the cap-stage incisor germs failed to enter the bell stage but triggered tremendous apoptosis in both epithelial and mesenchymal compartments. Compared to our previous reports using *Wnt1-cre*; *Rosa26R-Fgf8* mice in which incisor and molar germs arrested at the bud stage [23], our present study showed a special tooth agenesis due to both the maxillary and mandibular incisor regression at the later cap stage or early bell stage (Appendix A), expanding our putative notion that tooth agenesis could result from the failure not only in initiation but also in morphogenesis. The diminished *Shh* expression in the incisor epithelium was suggested to enhance apoptosis directly because Shh plays important roles in morphogenesis, cell proliferation, survival and differentiation during tooth development [22]. The diminished *Shh* transcription was implicated to result from the inactivation of *Runx2* in dental mesenchyme because *Runx2* is believed to activate *Fgf3* and *Fgf10* expression in dental mesenchyme, which, in turn, activated epithelial *Fgf4* and *Fgf9* expression to maintain *Shh* transcription [8,22]. On the other hand, since the suppressed *Runx2* expression also implicated the reduced FGF signaling in the incisor mesenchyme, the canonical Wnt signaling in the incisor mesenchyme was supposed to be elevated [8,13], which was supported by the enlarged *Lef1*-expressing domain. Series studies have demonstrated that the increased mesenchymal canonical Wnt activity, especially in the incisor mesenchyme, led to tooth agenesis by inhibiting tooth formation [13,24]. Therefore, the ectopic activation of *Fgf8* in the dental mesenchyme was suggested to result in incisor agenesis by suppressing epithelial *Shh* transcription and Runx2 expression but increasing mesenchymal canonical Wnt signaling. Actually, in the E13.5 sub-mandibular salivary glands of *Osr2-cre^KI^; Rosa26R-Fgf8* mice, the transcription of *Shh* and *Fgf10* in epithelial buds and ducts was also suppressed. Thus, how Fgf8 inhibits the expression of *Shh* and Runx2 in incisor germs still requires further investigation.

Compared to the incisor agenesis, the microdontia of *Osr2-cre^KI^*; *Rosa26R-Fgf8* molars was not so evident in the prenatal stage. The slightly inhibited epithelial *Shh* expression did not result in excessive apoptosis in the *Osr2-cre^KI^*; *Rosa26R-Fgf8* molar germs at the cap or bell stage. On the other hand, the mildly elevated mesenchymal Runx2, which implicated the increased FGF signaling, was suggested to promote cell proliferation [25]. Consistently, the obviously reduced Lef1 staining suggested the decreased canonical Wnt activity, and the activation of Fgfr1 implied an increased FGF signaling in E13.5 *Osr2-cre^KI^*; *Rosa26R-Fgf8* molar mesenchyme, implicating little impact on the prenatal odontogenic capacity of *Osr2-cre^KI^*; *Rosa26R-Fgf8* molars [25,26]. However, the persistent activation of *Fgf8* resulted in the delayed odontoblastic and premature ameloblastic differentiation of the sub-renal capsule-implanted *Osr2-cre^KI^*; *Rosa26R-Fgf8* molar germs, which impaired the following mineralization of enamel and dentin. The extended sub-renal culture indicated that the *Osr2-cre^KI^*; *Rosa26R-Fgf8* molar germs lost the typical tooth shape and structure but develop into a miniature tooth with the blurred cusps, which seemed to be mimicking microdontia [21]. It is strongly suggested that the persistent *Fgf8* expression impaired tooth root development by inactivating *Shh* transcription [22] and dentin mineralization by suppressing odontoblast differentiation and maturation [23,27].

The differential tooth phenotypes, agenesis and microdontia in *Osr2-cre^KI^; Rosa26R-Fgf8* incisors and molar germs, respectively, raised the question why dental mesenchymal-derived Fgf8 produced the different impacts on incisor and molar development. Our previous study also displayed the regressed maxillary and mandibular incisor germs, as well as the mildly impaired molar germs in *Osr2-cre^KI^*; *Ctnnb1^ex3f^* mice [28], which enforced the speculation that there was a potential difference between the incisor and molar. Actually, we indeed found that the expression of *Shh* and Runx2 was differential in the incisor and molar germs of *Osr2-cre^KI^*; *Rosa26R-Fgf8* mice. However, we still tend to interpret the differential tooth phenotypes by the differential pattern of *Osr2-cre^KI^* allele in incisor and molar mesenchyme. At E13.5, *Osr2-cre^KI^* was activated throughout the incisor mesenchyme (Appendix A), while it was activated only in the lingual mesenchyme of molar germs, which extended to entire molar mesenchyme at E15.5 [29]. The early and wider activation of *Fgf8* in incisor mesenchyme disabled the early events of tooth development [23], such as fate determination, cell proliferation and survival, leading to the more severe phenotype, tooth agenesis. In contrast, the late and partial activation of *Fgf8* in molar mesenchyme allowed tooth development into cytodifferentiation, which resulted in microdontia by disrupting cusp formation and mineralization. Such an interpretation integrated tooth agenesis and microdontia into a single pathogenesis which provides a novel perspective for the molecular mechanism of congenital tooth deformities.

Our study revealed that tooth agenesis could result from not only initiation failure but also the regression at late stage by excessive cell death. Furthermore, we indicated that the persistently active FGF signaling also led to microdontia with impaired root formation and mineralization, which disclosed the comprehensive effects of *Fgf8* on tooth development. This finding provides a novel etiological interpretation for tooth agenesis and microdontia by uniting two clinical manifestations into one pathogenesis, which would benefit the therapeutic strategy for tooth regeneration. Due to the recent and latest advances in the application potential of dental stem cells [30,31,32], our study also implicated that FGF signaling had to be finely tuned in the stem cells for tooth regeneration, and only a moderate activity would allow the normal differentiation and maturation of the stem cells. However, how the overactive FGF signaling impacts tooth root development, odontoblast differentiation and dentin mineralization still requires further investigation.

## 4. Materials and Methods

### 4.1. Mouse Lines

The *Osr2-cre^KI^*, *Rosa26R-Fgf8* and *Rosa26R-mT/mG* mice were genotyped as described previously [33]. All mice were housed and bred in the Specific Pathogenic Free System of the institute of Genome Engineered Animal Models for Human Diseases at Dalian Medical University. Morning of the day of vaginal plug detected in breeding females was designated as E0.5. Animals and procedures used in this study were approved by the Animal Care and Use Committee at Dalian Medical University (Protocol No. AEE18011).

### 4.2. Histological and Cryostat Section

The timed pregnant mice were treated by carbon dioxide and cervical dislocation for euthanasia. The dissected mouse embryonic heads were fixed in 4% paraformaldehyde (PFA) at 4 °C overnight, dehydrated through graded ethanol, embedded in paraffin and then sectioned in 10 μm for Masson staining [33]. The implanted molar germs from the sub-renal capsule culture were also subjected to the identical protocol for histological sections but stained by hematoxylin and eosin. For cryostat section, the heads were fixed in 4% PFA, dehydrated with 15% and 30% sugar solution overnight, embedded in O.C.T. and eventually sectioned into slices with a thickness of 10 μm. To ensure the penetrance of tooth phenotype and *Cre* pattern, the embryonic heads from at least three litters were examined for each stage.

### 4.3. BrdU Labeling Assay

For BrdU labeling assays, the pregnant mice were intraperitoneally injected with 10 mmol/L BrdU at a dosage of 5 mL/kg body weight at 30 min before sacrifice. The harvested embryonic heads were fixed in Carnoy’s fixative for 4 h, prepared for paraffin sections in 10 μm according to the above-mentioned protocols. The BrdU-labeled cells were detected using the Detection Kit II (Roche, Basel, Swiss) and counterstained with nuclear fast red solution. The density of BrdU-positive cells in incisor and molar germs was present by the numbers of BrdU labeling nuclei in the defined areas of dental epithelium and mesenchyme, respectively. The BrdU labeling assay was repeated in the samples from three different litters for each stage.

### 4.4. TUNEL Assay

Cell apoptosis within the epithelium and mesenchyme of incisor and molar germs was monitored by TUNEL assay, using the In Situ Cell Death Detection Kit, POD (Roche, Basel, Swiss) on 10 μm thick paraffin sections. Nuclei were counter-stained with DAPI. Cell density in incisor and molar germs was calculated by the numbers of TUNEL-positive cells in the defined areas of dental epithelium and mesenchyme, respectively. The TUNEL assay was performed in at least three repeats in the samples from different litters for each stage.

### 4.5. Statistical Assay

The number of BrdU- or TUNEL-labeled nuclei was counted with ImageJ version 1.51j8. The areas of dental epithelium and mesenchyme in incisor and molar germs were also quantified by ImageJ version 1.51j8. All quantitative data were expressed as means ± Standard Deviation from three independent repeats. Student’s *t*-test was used to determine the statistical significance of the differences, which were considered statistically significant at *p*-value < 0.05.

### 4.6. In Situ Hybridization

The molar germs of sub-renal culture were harvested in the ice-cold phosphate buffer solution treated with diethyl pyrocarbonate (DEPC). The DEPC-treated 4% PFA was used for fixation. After dehydration in DEPC-treated gradient ethanol, the molar germs were embedded in paraffin and sectioned in 10 μm for in situ hybridization, as previously described [33]. Antisense RNA probes for *Shh*, *Ectodin*, *Amelogenin* and *Dspp* were generated as reported previously [28,29]. Eosin was used for counter-staining. For *Shh* and *Ectodin* probes, the samples from three different litters were examined. For the probes of *Amelogenin* and *Dspp*, three samples were examined for each stage.

### 4.7. Immunohistochemistry and Immunofluorescence

Immunocytochemistry was performed as described previously [33]. Primary antibodies used for immunocytochemistry were phosphorylated-Smad 1/5/8 (p-Smad1/5/8, 1:200 dilution, Cell Signaling Technology, Boston, USA), phosphorylated Erk1/2 (p-Erk1/2, 1:200 dilution, Abcam, Cambridge, UK), Lef1 (1:1000 dilution, Abcam, Cambridge, UK) and Fgfr1 (1:400 dilution, Cell Signaling Technology). The horseradish peroxidase (HRP)-conjugated anti-rabbit/mouse IgG was used as the secondary antibody, and the DAB (Maixin Ltd. Shenzhen, China) for color development. After color development, the section was counter-stained with hematoxylin. The primary antibody against Runx2 (1:1000 dilution, Abcam, Cambridge, UK) was applied for immunofluorescence with goat anti-rabbit IgG (Alexa Flour 555, Abcam, Cambridge, UK) as secondary antibody and DAPI counterstaining. Each immunohistochemical and immunofluorescence consequence was triply repeated with the samples from different litters.

### 4.8. X-ray Plain Film for Sub-Renal Capsule Implanted Molar Germs

The E17.5 molar germs were isolated and transplanted under the kidney capsule of adult host mice for 3-day, 5-day, 3-week or 6-week culture, respectively, as previously described [24]. The grafted molar germs were collected and fixed in 4% PFA. The X-ray plain films of the implanted molars were obtained by using X-ray radiography (Faxitron MX-20DC12, Faxitron Bioptics, Boston, MA, USA). For each period of sub-renal culture, at least three molar germs were implanted and examined by X-ray radiography.

## 5. Conclusions

In summary, our study reported that ectopic activation of *Fgf8* in dental mesenchyme by *Osr2-cre^KI^* resulted in incisor agenesis and molar microdontia. The incisor agenesis in *Osr2-cre^KI^*; *Rosa26R-Fgf8* mice was attributed to the tremendous apoptosis at the early bell stage, which resulted from the suppressed expression of *Shh* and Runx2 and the increased mesenchymal Wnt canonical signaling. In contrast, although affected a little prenatally, *Osr2-cre^KI^*; *Rosa26R-Fgf8* molars only formed a miniature tooth with impaired odontoblast and ameloblast differentiation, and mineralization after sub-renal culture, which resembled microdontia. The severe phenotype in the *Osr2-cre^KI^*; *Rosa26R-Fgf8* incisor was most likely attributed to the early and wider activation of *Osr2-cre^KI^* in the developing incisor mesenchyme. Thus, it implicated that tooth agenesis and microdontia could result from a single pathogenesis, which displayed the differentially spatial–temporal pattern in tooth development.

## Figures and Tables

**Figure 1 ijms-25-07045-f001:**
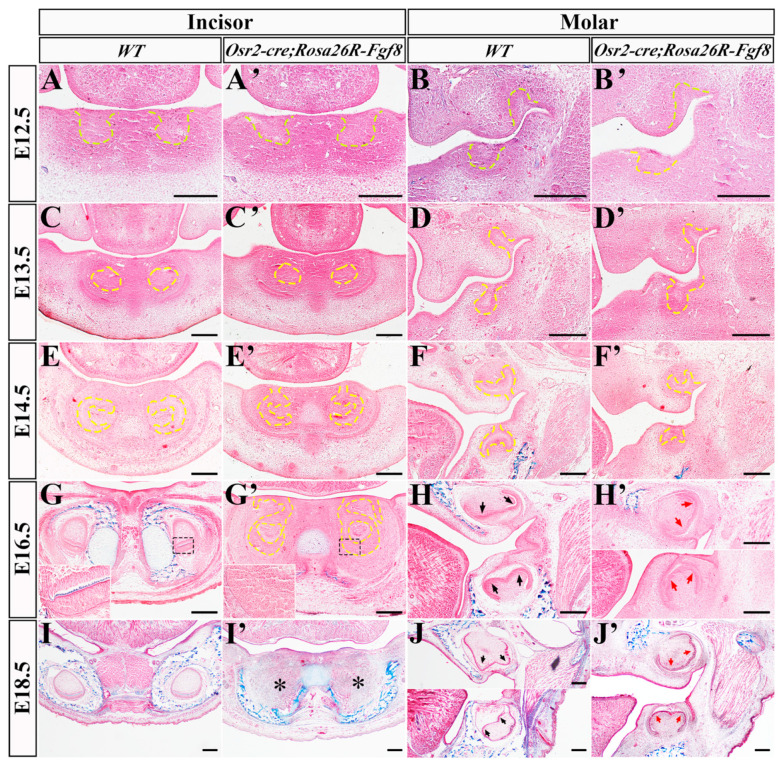
Histological features of the *Osr2-cre^KI^*; *Rosa26R-Fgf8* incisor and molar germs. (**A**–**J**) Masson staining showed WT E12.5 incisor (**A**) and molar germs (**B**), E13.5 incisor (**C**) and molar germs (**D**), E14.5 incisor (**E**) and molar germs (**F**), E16.5 incisor (**G**) and molar germs (**H**), and E18.5 incisor (**I**) and molar germs (**J**). (**A′**–**J′**) Masson staining showed *Osr2-cre^KI^*; *Rosa26R-Fgf8* E12.5 incisor (**A′**) and molar germs (**B′**), E13.5 incisor (**C′**) and molar germs (**D′**), E14.5 incisor (**E′**) and molar germs (**F′**), E16.5 incisor (**G′**) and molar germs (**H′**), and E18.5 incisor (**I′**) and molar germs (**J′**). The yellow dashed lines delineate the contour of dental epithelium, and asterisks indicate degenerated incisor germs. Black arrows in (**H**,**J**) indicate the secondary EKs of WT molar germs, while red arrows in (**H′**,**J′**) indicate the blurred secondary EKs of *Osr2-cre^KI^*; *Rosa26R-Fgf8* molar germs. Scale bar, 200 μm.

**Figure 2 ijms-25-07045-f002:**
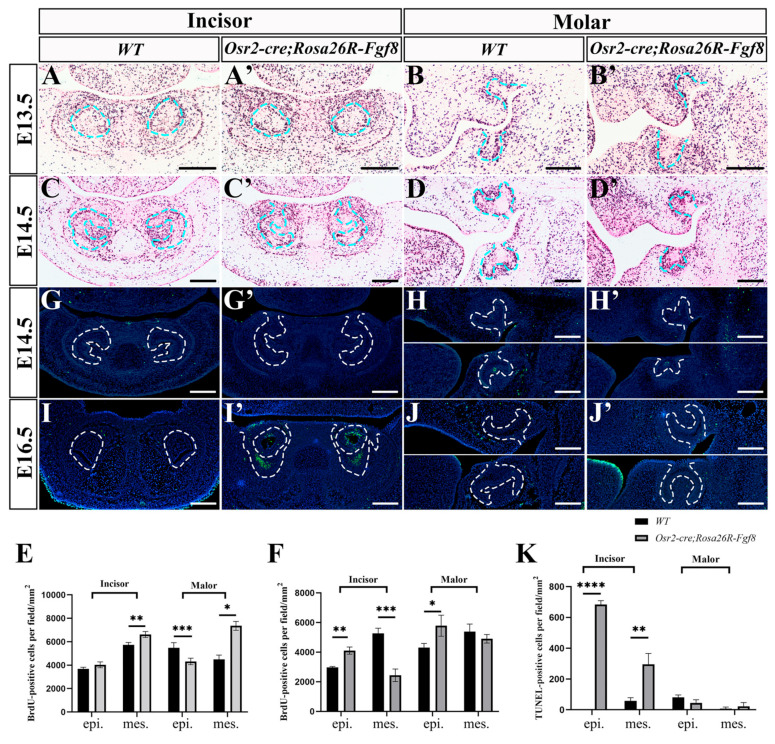
The cell proliferation and apoptosis assays in *Osr2-cre^KI^*; *Rosa26R-Fgf8* incisor and molar germs. (**A**–**D**) BrdU-positive cells in the E13.5 WT incisor (**A**) and molar germs (**B**), and the E14.5 incisor (**C**) and molar germs (**D**). (**A′**–**D′**) BrdU-positive cells in E13.5 *Osr2-cre^KI^*; *Rosa26R-Fgf8* incisor (**A′**) and molar (**B′**) germs, and E14.5 incisor (**C′**) and molar (**D′**) germs. (**E**) The densities of BrdU-positive cells in the E13.5 WT (3675.14 ± 120.48 cells/mm^2^) vs. *Osr2-cre^KI^*; *Rosa26R-Fgf8* incisor epithelium (4026.11 ± 203.43 cells/mm2, *p* = 0.1037); in the E13.5 WT (5739.88 ± 162.88 cells/mm^2^) vs. *Osr2-cre^KI^*; *Rosa26R-Fgf8* incisor mesenchyme (6614.66 ± 208.83 cells/mm^2^, *p* = 0.0095); in the E13.5 WT (5477.38 ± 360.16 cells/mm^2^) vs. *Osr2-cre^KI^*; *Rosa26R-Fgf8* molar epithelium (4314.54 ± 228.91 cells/mm^2^, *p* = 0.0007); and in the E13.5 WT (4492.09 ± 300.34 cells/mm^2^) vs. *Osr2-cre^KI^*; *Rosa26R-Fgf8* molar mesenchyme (7359.18 ± 308.11 cells/mm^2^, *p* = 0.0182). (**F**) The densities of BrdU-positive cells in the E14.5 WT (2970.73 ± 48.05 cells/mm^2^) vs. *Osr2-cre^KI^*; *Rosa26R-Fgf8* incisor epithelium (4108.11 ± 199.80 cells/mm^2^, *p* = 0.0014); in the E14.5 WT (5265.59 ± 287.20 cells/mm^2^) vs. *Osr2-cre^KI^*; *Rosa26R-Fgf8* incisor mesenchyme (2434.35 ± 348.76 cells/mm^2^, *p* = 0.0009); in the E14.5 WT (4300.66 ± 234.13 cells/mm^2^) vs. *Osr2-cre^KI^*; *Rosa26R-Fgf8* molar epithelium (5787.29 ± 581.55 cells/mm^2^, *p* = 0.0285); and in the E14.5 WT (5390.52 ± 414.84 cells/mm^2^) vs. *Osr2-cre^KI^*; *Rosa26R-Fgf8* molar mesenchyme (4904.49 ± 232.50 cells/mm^2^, *p* = 0.2219). (**G**–**J**) TUNEL assay indicated the cell death in the E14.5 WT incisor (**G**) and molar germs (**H**), and the E16.5 WT incisor (**I**) and molar germs (**J**). (**G′**–**J′**) TUNEL assay indicated the cell death in the E14.5 *Osr2-cre^KI^; Rosa26R-Fgf8* incisor (**G′**) and molar germs (**H′**), and in the E16.5 *Osr2-cre^KI^*; *Rosa26R-Fgf8* incisor (**I′**) and molar germs (**J′**). (**K**) The densities of TUNEL-positive cells in the E16.5 WT (0 ± 0 cells/mm^2^) vs. *Osr2-cre^KI^*; *Rosa26R-Fgf8* incisor epithelium (683.39 ± 21.02 cells/mm^2^, *p* < 0.0001); in the E16.5 WT (58.96 ± 16.31 cells/mm^2^) vs. *Osr2-cre^KI^*; *Rosa26R-Fgf8* incisor mesenchyme (296.01 ± 57.36 cells/mm^2^, *p* = 0.0049); in the E16.5 WT (81.85 ± 12.46 cells/mm^2^ vs. *Osr2-cre^KI^*; *Rosa26R-Fgf8* molar epithelium (45.92 ± 16.30 cells/mm^2^, *p* = 0.0685); and in the E16.5 WT (6.72 ± 9.51cells/mm^2^) vs. *Osr2-cre^KI^*; *Rosa26R-Fgf8* molar mesenchyme (23.16 ± 20.13 cells/mm^2^, *p* = 0.3555). Blue and white dotted lines indicated the boundary between the dental epithelium and mesenchyme. (epi, epithelium; mes, mesenchyme; * *p* < 0.05; ** *p* < 0.01; *** *p* < 0.001; **** *p* < 0.0001; scale bar, 200 μm).

**Figure 3 ijms-25-07045-f003:**
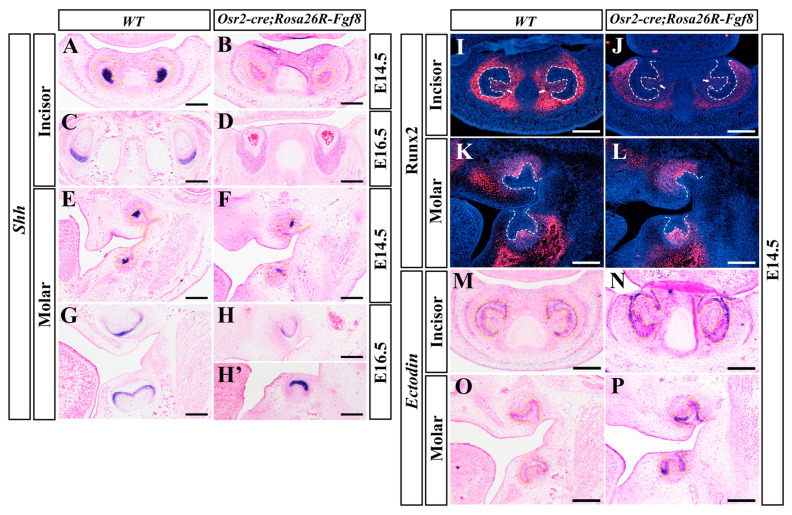
The pattern of *Shh*, Runx2 and *Ectodin* in the developing *Osr2-cre^KI^*; *Rosa26R-Fgf8* incisor and molar germs. (**A**–**D**) The in situ hybridization for *Shh* transcription in the E14.5 WT (**A**) and *Osr2-cre^KI^*; *Rosa26R-Fgf8* incisor germs (**B**), as well as in the E16.5 WT (**C**) and *Osr2-cre^KI^*; *Rosa26R-Fgf8* incisor germs (**D**). (**E**–**H**,**H′**) In situ hybridization showed *Shh* transcription in the E14.5 WT (**E**) and *Osr2-cre^KI^*; *Rosa26R-Fgf8* molar germs (**F**), as well as in the E16.5 WT (**G**) and *Osr2-cre^KI^*; *Rosa26R-Fgf8* molar germs (**H**,**H′**). (**I**–**L**) The immunofluorescence of Runx2 in the E14.5 WT (**I**) and *Osr2-cre^KI^; Rosa26R-Fgf8* incisor germs (**J**), as well as in the E14.5 WT (**K**) and *Osr2-cre^KI^*; *Rosa26R-Fgf8* molar germs (**L**). (**M**–**P**) In situ hybridization showed *Ectodin* transcription in the E14.5 WT (**M**) and *Osr2-cre^KI^*; *Rosa26R-Fgf8* incisor germs (**N**), as well as in the E14.5 WT (**O**) and *Osr2-cre^KI^*; *Rosa26R-Fgf8* molar germs (**P**). The yellow and white dashed lines mark contoured enamel organs in incisor and molar germs. Scale bar, 200 μm.

**Figure 4 ijms-25-07045-f004:**
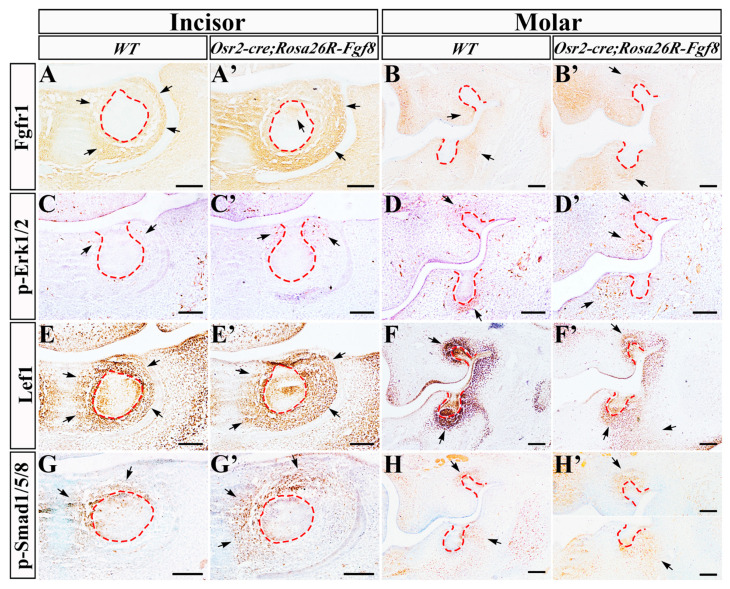
The altered signaling pathways in *Osr2-cre^KI^*; *Rosa26R-Fgf8* tooth germs. The Fgfr1 immunostaining in the E13.5 WT incisor (**A**) and molar germs (**B**), and the E13.5 *Osr2-cre^KI^*; *Rosa26R-Fgf8* incisor (**A′**) and molar germs (**B′**). The immunostaining of p-Erk1/2 in the E13.5 WT incisor (**C**) and molar germs (**D**), and the E13.5 *Osr2-cre^KI^*; *Rosa26R-Fgf8*incisor (**C′**) and molar germs (**D′**). The Lef1 immunostaining in the E13.5 WT incisor (**E**) and molar germs (**F**), and the E13.5 *Osr2-cre^KI^*; *Rosa26R-Fgf8* incisor (**E′**) and molar germs (**F′**). The immunostaining of p-Smad1/5/8 in the E13.5 WT incisor (**G**) and molar germs (**H**), and the E13.5 *Osr2-cre^KI^*; *Rosa26R-Fgf8* incisor (**G′**) and molar germs (**H′**). The red dashed lines delineate the contour of dental epithelium, and the black arrows point to the positive staining in incisor and molar germs. Scale bar, 200 μm.

**Figure 5 ijms-25-07045-f005:**
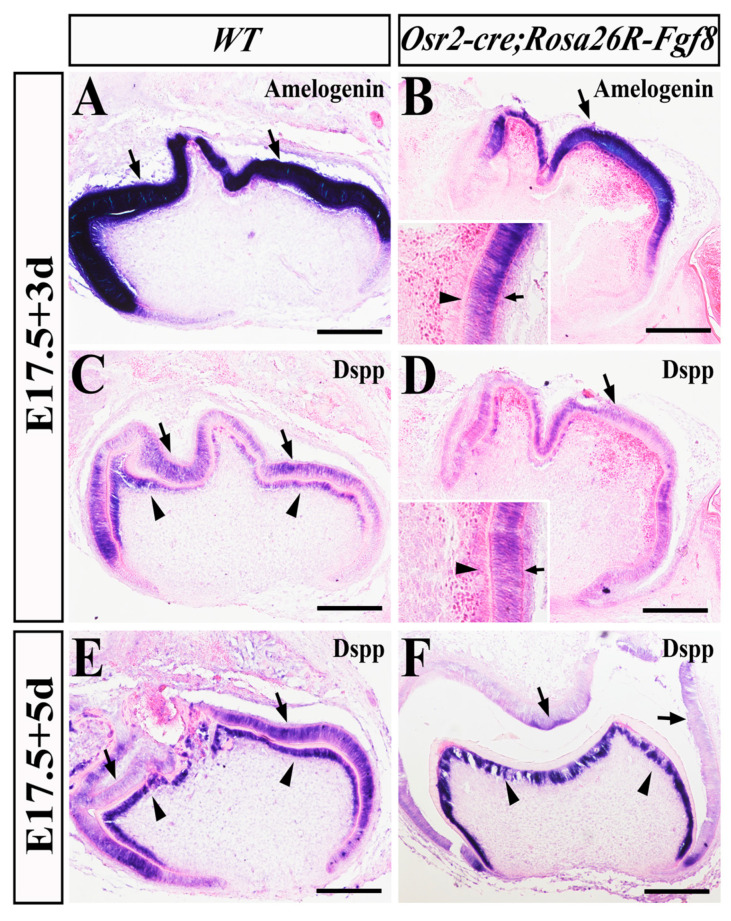
In situ hybridization for *Amelogenin* and *Dspp* expression in implanted *Osr2-cre^KI^*; *Rosa26R-Fgf8* molar germs. After 3 days of sub-renal culture of E17.5 molar germs, in situ hybridization showed the *Amelogenin* expression in the WT (**A**) and *Osr2-cre^KI^*; *Rosa26R-Fgf8* molar germs (**B**). After the 3-day sub-renal culture of E17.5 molar germs, in situ hybridization showed the *Dspp* expression in the WT (**C**) and *Osr2-cre^KI^*; *Rosa26R-Fgf8* molar germs (**D**). After 5 days of sub-renal culture, in situ hybridization showed the *Dspp* expression in the E17.5 WT (**E**) and *Osr2-cre^KI^*; *Rosa26R-Fgf8* molar germs (**F**). The arrows indicate ameloblasts, while the arrowhead points to odontoblasts. Scale bar, 200 μm.

**Figure 6 ijms-25-07045-f006:**
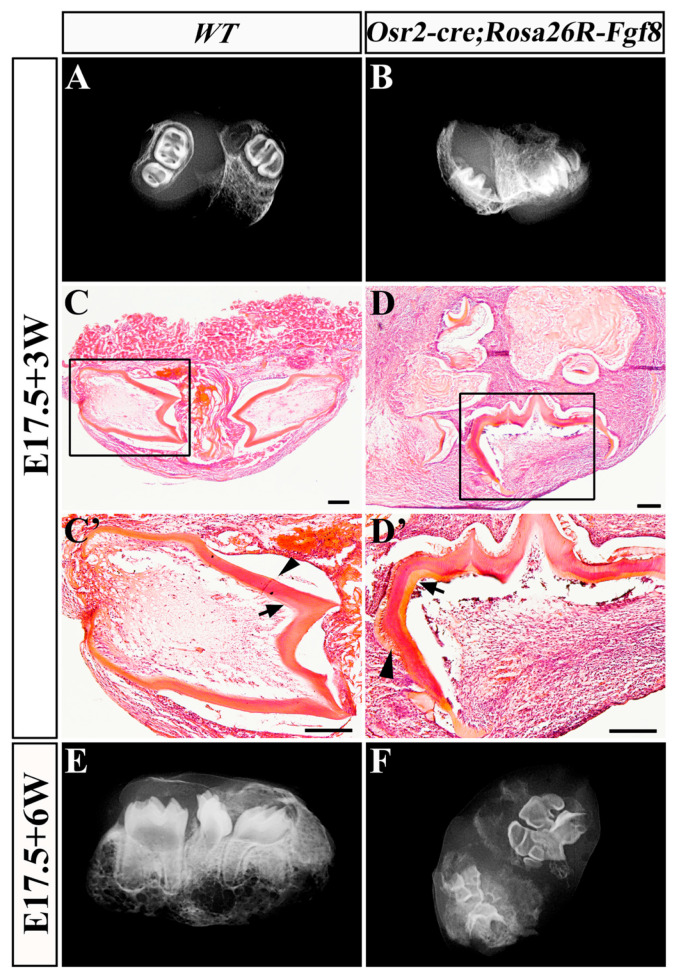
Structure and histology of the grafted *Osr2-cre^KI^*; *Rosa26R-Fgf8* molar germs. After 3 weeks of sub-renal culture, plain X-ray images showed the tooth structure of the grafted E17.5 WT (**A**) and *Osr2-cre^KI^*; *Rosa26R-Fgf8* molar germs (**B**). HE staining shows the histological features of the E17.5 WT (**C**) and *Osr2-cre^KI^; Rosa26R-Fgf8* (**D**) molar grafts after 3 weeks of sub-renal culture. (**C′**,**D′**) are the magnified views in the boxes of (**C**,**D**), respectively. After 6 weeks of sub-renal culture, plain X-ray images showed the tooth morphology of the grafted E17.5 WT (**E**) and *Osr2-cre^KI^*; *Rosa26R-Fgf8* molar germs (**F**). The black arrows point to the predentin, and the arrowhead points to the dentin in molar grafts. Scale bar, 100 μm.

## Data Availability

The data of this study are available from the corresponding authors upon the reasonable request.

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
