# Peer review of "Ectopic Activation of Fgf8 in Dental Mesenchyme Causes Incisor Agenesis and Molar Microdontia"

_ijms, 2024, doi:10.3390/ijms25137045_

Round 1
Reviewer 1 Report
Comments and Suggestions for Authors
Dear Authors,
Congratulations of the job you have done and presented in this manuscript. I believe that your work is significant to the field and might be of high interest for the general reader. However, for the moment, I cannot recommend publication of this paper in a such prestigious journal. please see the attachment.

Minor revisions are required
Author Response
Since the comments were labeled in the manuscript in a PDF format, we answer the comments point to point in the labled manuscript. Please see the attachment.

Reviewer 2 Report
Comments and Suggestions for Authors
Dear Authors,
you made a great work! However, some improvements are suggested before acceptance.

Author Response
Comment 1: In the Results section, are the figures included protected by copyright? (Fig. 1 2 3 4 5 6). Do you think it is possible, without reducing the number of paragraphs, to summarize this section which I believe to be complete, but quite didactic and narrative in some sections, such as: 2.3 and 2.6.
Response: All the images and figures are the original work in our lab. There is no copyright conflict in these images and figures. We also added the summary sentences for Results 2.3 and 2.6 to make it easier to be understand by readers.
Comments 2: Bibliography should be formatted respecting the journal’s requirements and no improper citations are evidenced.
Response: Since we added new references in the revised manuscript as reviewer required, we double checked and confirmed the format of the references.
Reviewer 3 Report
Comments and Suggestions for Authors
This is a relevant study that investigated the potential factors and mechanisms related to dental agenesis and microdontia.
Some considerations:
1) I suggest that the Authors better highlight the objective of the study in “Abstract”;
2) Add sufficiently detailed information in “Materials and Methods” to allow for reproducibility and clarity of the study. Additionally, add more precise information about animal procedures (such as number of animals, euthanasia);
3) The results could be discussed more extensively by the Authors, especially the molecular and cellular aspects of odontogenesis and genetic, metabolic and environmental factors that influence and regulate the formation and development of dental tissues, including dental mineralization, as well as the association between conditions oral and systemic.
Author Response
This is a relevant study that investigated the potential factors and mechanisms related to dental agenesis and microdontia.
Some considerations:
1) I suggest that the Authors better highlight the objective of the study in “Abstract”;
Response: Thank reviewer's suggestion. We rewrite the Abstract and hightlight the object of the study.
2) Add sufficiently detailed information in “Materials and Methods” to allow for reproducibility and clarity of the study. Additionally, add more precise information about animal procedures (such as number of animals, euthanasia);
Reponse:We added the exact precedures and mpuse numbers used in each experiment as reviewer required.
3) The results could be discussed more extensively by the Authors, especially the molecular and cellular aspects of odontogenesis and genetic, metabolic and environmental factors that influence and regulate the formation and development of dental tissues, including dental mineralization, as well as the association between conditions oral and systemic.
Reponse: We added new paragraphs and new references to extensively discuss the content mentioned by reviewer. We also added a conclusion section to summarize the Results and Discussion.
Round 2
Reviewer 1 Report
Comments and Suggestions for Authors
Dear Authors, I believe that you took advantage of the reviewers comments and improved a lot the paper. I have no further comments.
Reviewer 3 Report
Comments and Suggestions for Authors
The adjustments improved and complemented the manuscript.